# Lactic Acid Fermentation of *Chlorella vulgaris* to Improve the Aroma of New Microalgae-Based Foods: Impact of Composition and Bacterial Growth on the Volatile Fraction

**DOI:** 10.3390/foods14091511

**Published:** 2025-04-26

**Authors:** Caterina Nicolotti, Martina Cirlini, Lorenzo Del Vecchio, Jasmine Hadj Saadoun, Valentina Bernini, Monica Gatti, Benedetta Bottari, Francesco Martelli

**Affiliations:** Department of Food and Drug, University of Parma, Viale delle Scienze 49/A, 43124 Parma, Italy; caterina.nicolotti@unipr.it (C.N.); martina.cirlini@unipr.it (M.C.); lorenzo.delvecchio@unipr.it (L.D.V.); jasmine.hadjsaadoun@unipr.it (J.H.S.); valentina.bernini@unipr.it (V.B.); monica.gatti@unipr.it (M.G.)

**Keywords:** fermentation, flavor improvement, microalgae, volatile compounds

## Abstract

The consumption of microalgae-based foods is growing due to their exceptional nutritional benefits and sustainable cultivation. However, their strong off-flavors and odors hinder their incorporation into food products. Lactic acid fermentation, a traditional method known for modifying bioactive and aromatic compounds, may address these challenges. This study aims to evaluate the impact of lactic acid fermentation on the aromatic profiles of four distinct *Chlorella vulgaris* biomasses, each varying in protein, carbohydrate, lipid, and pigment content. Six lactic acid bacteria (LAB) strains, *Lacticaseibacillus casei*, *Lcb. paracasei*, *Lcb. rhamnosus*, *Lactiplantibacillus plantarum*, *Lactobacillus delbrueckii subsp. bulgaricus*, and *Leuconostoc citreum*, were used for fermentation. All biomasses supported LAB growth, and their volatile profiles were analyzed via HS-SPME-GC-MS, revealing significant variability. Fermentation notably reduced concentrations of compounds responsible for off-flavors, such as aldehydes. Specifically, hexanal, associated with a green and leafy aroma, was significantly decreased. *Lcb. paracasei* UPCCO 2333 showed the most effective modulation of the volatile profile in *Chlorella vulgaris*, significantly reducing undesirable compounds, such as aldehydes, ketones, pyrazines, and terpenes, while enhancing ester production. These results highlight lactic acid fermentation as an effective method to improve the sensory characteristics of *C. vulgaris* biomasses, enabling their broader use in innovative, nutritionally rich food products.

## 1. Introduction

The consumption of microalgal and cyanobacterial biomasses as nutraceuticals is increasingly gaining interest because of their valuable composition in proteins, vitamins, minerals, polyunsaturated fatty acids, and bioactive compounds [1,2]. At the same time, their use in food formulation is growing because of technological, nutritional, and coloring properties [3]. Furthermore, because of their composition, microalgae are considered one of the future foods as single-cell proteins, and many researchers and industries are investing in their development [4]. *Chlorella vulgaris* is the second most cultivated microalgae, after the cyanobacterium *Arthrospira platensis* [5]. Since *Chlorella* was widely consumed before May 1997, the European Food Safety Authority (EFSA) does not classify it as a Novel Food; hence, there are no restrictions on its usage or marketing that apply to other algae [6]. This Chlorophyta has a good concentration of proteins (50–60% dry weight); it has good quality essential amino acids and carbohydrates; and it can reach a lipid content of 5–40% dw, consisting of glycolipid waxes, phospholipids, and fatty acids (palmitic acid, stearic acid, palmitoleic acid, and oleic acid) [7]. Furthermore, it can be easily cultivated in open ponds or photobioreactors and is fast in growth [8]. One of the biggest challenges in incentivizing consumption and formulating food with microalgae is related to the unpleasant flavor and smell of the biomass. In fact, their volatile compounds strongly influence the flavor and aroma of microalgae giving them a taste described as muddy and fishy [9]. *Chlorella vulgaris* has been reported to present an odor profile dominated by grassy and vegetable aromas, dominated by a high presence of aldehydes and ketones [10]. The production of volatile organic compounds (VOCs) by microalgae is subject to variations but can also be influenced by light, salt concentration, carbon and nitrogen sources, and other factors. Although *Chlorella*’s VOCs have been the subject of numerous investigations, nothing is known about how they are produced in heterotrophic or phototrophic environments. Generally, the presence of chlorophyll is related to the presence of the biomasses’ bad taste. For this reason, the chlorophyll content of some microalgal biomass can be reduced by cultivation in heterotrophic conditions or by selecting low chlorophyll strains [11].

At the same time, lactic acid fermentation has been used by humans for many centuries not only as a preservation technique but also to improve and change the aroma of several matrices [12,13]. A biological process that, in addition to increasing the safety and healthiness of a product, reduces unpleasant smell and taste may represent an important tool for treatment of algal biomass to facilitate the inclusion in complex foods with reduced off-flavors or with improved aroma. Lactic acid bacteria (LAB) metabolism leads not only to fast acidification of the substrate and rapid consumption of fermentable sugars, with a competitive advantage in the use of LAB in nutrient-rich environments, but also to the production of volatile compounds belonging to different chemical classes, such as alcohols, aldehydes, ketones, acids, esters, and sulfur compounds. These compounds mainly derive from the catabolism of citrate and the degradation of proteins and lipids. The formation of VOCs is a complex process in which precursors are initially generated and subsequently converted into aromatic compounds [14]. Due to their composition, characterized by a high percentage of amino acids and carbohydrates, microalgae are suitable matrices for lactic acid fermentation. Numerous applications have been investigated in the relationship between LAB and microalgae, particularly through fermentation. Algal fermentation has been used to improve bioactivities, with positive impacts on the growth of probiotics, aromatic improvement, and functional foods production [15,16,17].

The present study aimed to identify characteristic aroma compounds of four different commercial *C. vulgaris*, with different compositions, cultivated in heterotrophic or phototrophic conditions, through Gas Chromatography–Mass Spectrometry (GC-MS) analysis using a Headspace Solid-Phase Microextraction (HS-SPME). With the aim of improving the aromatic profile and enhancing the sensory appeal of the four biomasses, solid-state lactic acid fermentation was applied as a strategy to modulate their flavor and increase their suitability for food applications. Six different species of LAB, with the Qualified Presumption of Safety (QPS) status, belonging to the Lactobacillaceae family (*Lacticaseibacillus rhamnosus*, *Lactobacillus delbrueckii bulgaricus*, *Lacticaseibacillus casei*, *Lacticaseibacillus paracasei*, *Leuconostoc citreum*, and *Lactiplantibacillus plantarum*) were used as starters of the microalgal biomass [18]. The selection of starter cultures to develop novel fermented microalgal biomass with improved aroma could play an important role in increasing the commercialization and consumption of this kind of single-cell protein [19].

## 2. Materials and Methods

### 2.1. Microalgal Biomasses and Bacterial Strains Used for Fermentation

Commercial *Chlorella vulgaris* biomasses were used for the experiments. The biomasses were purchased from Allmicroalgae (Leira, Portugal); in particular, *C. vulgaris* “Smooth” (CHL1), *C. vulgaris* “Premium” (CHL2), *C. vulgaris* “Honey” (CHL3), and *C. vulgaris* “White” (CHL4) were used. Nutritional compositions of the commercial biomasses are shown in Table 1. Six QPS LAB strains, belonging to the UPCCO (University of Parma Culture Collection, Parma, Italy), were used to ferment microalgal biomasses. The strains were previously isolated from different food matrixes; identified by 16S rRNA sequencing (Table 2); and maintained at −80 °C in de Man, Rogosa, and Sharpe medium (MRS) (Oxoid, Basingstoke, UK), complemented with 12.5% glycerol (*v*/*v*).

### 2.2. Set Up of Fermentations

LAB strains were revitalized twice in MRS broth (Oxoid) with an inoculum of 3% (*v*/*v*) and incubated for 16 h at either 37 °C or 30 °C under aerobic conditions. Subsequently, each revitalized strain was inoculated into fresh MRS broth (3% *v*/*v*) and incubated for 15 h at the microorganism’s optimal growth temperature, achieving a bacterial concentration of 9 Log CFU/mL. The cultured cells were then harvested via centrifugation at 10,000 rpm for 10 min at 4 °C, washed twice with Ringer solution (Oxoid, Milan, Italy), and resuspended in sterile bi-distilled water. Each microalgal biomass was rehydrated with 75% sterile water and individually inoculated with each bacterial suspension to achieve a final bacterial concentration of 7 Log CFU/g in the inoculated matrix. Microbial concentrations were assessed immediately after inoculation (T0) and after 24 (T1), 48 (T2), and 72 h (T3) of fermentation using plate counts. Serial dilutions of the samples were prepared in Ringer solution (Oxoid) and plated onto MRS agar, followed by incubation for 48 h under aerobic conditions at the strains’ optimal growth temperature of 37 °C or 30 °C. The pH of all samples was measured at each fermentation stage using a Mettler Toledo pH meter (Greifensee, Switzerland). Fermentations were performed in duplicate, with analyses conducted in duplicate for each sampling point. Results were reported as mean values ± standard deviations of Log CFU/g. After fermentation, the biomasses were lyophilized for 48 h using a Freeze Dryer Lio-5P (5Pascal, Milan, Italy) and stored at −20 °C.

### 2.3. HS-SPME/GC-MS Analysis

Volatile compound (VOC) determination was performed by means of HS-SPME/GC-MS technique. All the parameters applied for these analyses were the same described by Martelli and collaborators in 2020 [17], both in term of sample preparation and extraction, as in terms of gas-chromatographic separation and mass spectrometric detection. Four different replicates for each sample were analyzed. All the identified VOCs were semi-quantified by using a reference component (Toluene), and results were reported as ng/g.

### 2.4. Statistical Analysis

Graphpad Prism (Version 8, San Diego, CA, USA) was used to create graphs and to develop statistical analyses. A two-way ANOVA was performed (*p* < 0.05) to evaluate significant differences in each LAB species’ ability to ferment microalgal biomasses, as well as the impact of fermentation on variations in volatile compound classes. The VOCs classes data were normalized and Log-transformed [Log10 (ng/g)].

Furthermore, to better highlight the analogies and/or differences among *Chlorella* samples as between the fermented products, principal component analysis (PCA) was applied, using R v4.3.2 environment [20], considering as variables the concentrations of the different VOCs classes identified in the analyzed samples.

## 3. Results

### 3.1. Chlorella Fermentation

Four different *C. vulgaris* biomasses were fermented with six LAB strains belonging to six different species: *Lacticaseibacillus rhamnosus* UPCCO 1473, *Lactobacillus delbrueckii bulgaricus* UPCCO 1932, *Lacticaseibacillus casei* UPCCO 2240, *Lacticaseibacillus paracasei* UPCCO 2333, *Leuconostoc citreum* UPCCO 4516, and *Lactiplantibacillus plantarum* UPCCO 4932. Growth ability of LABs strains is displayed in Table 3, and curves are shown in Figure 1.

The microbial growth was evaluated after 24, 48, and 72 h of incubation at the optimal temperature of the tested species (37 or 30 °C). Generally, all the strains showed good growth ability in all biomasses. Overall, a higher LAB growth was observed in CHL2, which was the *Chlorella* biomass with the higher protein concentration (55%), grown in phototrophic conditions. While fermenting CHL2, all the strains tested, except UPCCO 1932, increased their concentration by more than 2 Log CFU/g (Figure 2a).

The biomass that showed the lowest fermentability is CHL4. This heterotrophically grown *C. vulgaris* biomass is characterized by a high carbohydrate concentration (55%) and a fair amount of proteins (32.5%). The starting pH of CLH4 was lower compared to the other biomasses (CHL4= 4.9 ± 0.01; CHL1 = 6.06 ± 0.01; CHL2 = 6.73 ± 0.01 and CHL3 = 6.11 ± 0.01). This low starting pH could be the reason for the lower LAB growth rate until the end of the fermentation.

The best growth performance was observed for *Lcb. casei* UPCCO 2240, with a ΔT Log UFC/g increase of 2.71 ± 0.06 Log CFU/g (Table 3). Despite this increase, the strain was not the best-performing strain in acidifying the biomass. The final pH of the 72 h fermented CHL2 biomass was 5.55 ± 0.04, proving an acidification of 1.17 points. A better performance in acidifying the biomass has been seen while fermenting CHL1 and CHL2, reaching a pH below 4.5 already after 24 h of fermentation (4.24 and 4.25, respectively). The strain that had the lowest growth rate was *Leu. citreum* UPCCO 4516, especially in CHL3 and CHL4. The same strain did not show a strong acidification ability in any of the selected biomasses (Figure 2b).

*Lcb. rhamnosus* UPCCO 1473 was able to grow well on *C. vulgaris* biomasses with an increase of more than 2 Log CFU/ g (CHL2 ΔT = 2.17 ± 0.01 Log CFU/g and CHL3 ΔT = 2.05 ± 0.08 Log CFU/g). The strain showed a good acidification ability while fermenting CHL1 and CHL3.

Also, *L. delbrueckii bulgaricus* UPCCO 1932 showed a ΔT Log UFC/g higher than two Log CFU/g in two different biomasses (CHL1 ΔT = 2.27 ± 0.05 Log CFU/g and CHL3 ΔT = 2.23 ± 0.03 Log CFU/g) (Figure 2a). The strain showed a good acidification performance in this type of biomass. All the *L. delbrueckii bulgaricus* fermented biomasses, with the exemption of CHL2, showed a pH below 4.5 after 72 h of fermentation (Figure 2b).

*Lcb. paracasei* UPCCO 2333 was able to grow well on the selected microalgal biomass (CHL1 ΔT Log UFC/g = 1.04 ± 0.2; CHL2 ΔT Log UFC/g = 1.38 ± 0.04; CHL3 ΔT Log UFC/g = 2.16 ± 0.37; CHL4 ΔT Log UFC/g = 0.90 ± 0.05). This strain demonstrated a good acidification capacity while fermenting CHL3 and CHL4 (4.26 and 4.01 of pH, respectively). On the other hand, the strain showing the best acidification ability was *Lpb. plantarum* UPCCO 4932. All the biomasses fermented by *Lpb. plantarum* had a pH lower than 4.5 after 72 h of fermentation. The highest acidification ability has been observed while fermenting CHL2, causing an acidification of the biomass of 2.5. A ΔT Log UFC/g higher than 2 Log CFU/g has been seen only after fermentation of CHL2.

### 3.2. Volatile Profile Characterization of C. vulgaris and Changes in Volatile Components After Fermentation

The volatile profile characterization of *C. vulgaris* biomasses led us to identify a total of 106 different components that pertained to several chemical classes, such as acids, alcohols, aldehydes, esters, furans, aromatic hydrocarbons, hydrocarbons, ketones, sulfur containing compounds, pyrazines, and terpenes. All the identified VOCs have been semi-quantified, and their concentrations (ng/g) are reported in Appendix A. Given the complexity of the volatile profile in the analyzed samples, a linear dimensionality reduction technique was applied to better identify potential similarities and differences among *C. vulgaris* strains, particularly in relation to the impact of fermentation on the aromatic fraction.

Results reported herein show evidence of great variability in the volatile compounds in the different *C.* biomasses analyzed (Figure 3).

The PCA biplot (Figure 3a) graph depicts the formation of three different clusters. CHL1 and CHL3 are grouped in a single cluster and are characterized by a prevalence of aromatic hydrocarbons, acids, and pyrazines. CHL4 is characterized by a different quantity of ketones, alcohols, esters and aldehydes. CHL2, the biomass grown in autotrophic conditions, is characterized by a different content of hydrocarbons, furans, and terpenes. The analysis has pictured alcohols, acids, aldehydes, ketones, and pyrazines as the most dominant classes in the *C. vulgaris* biomasses (Figure 3b). In particular, the most dominant class of VOCs was aldehydes, with a strong presence of hexanal. Furans have been found only in CHL1 and CHL2. A low amount of esters has been detected in CHL2, CHL3, and CHL4.

The CHL4 biomass, commercially known as *Chlorella* White, is the biomass with the highest amount of alcohols, aldehydes, and ketones.

Figure 4 represents the results obtained by the PCA analysis applied to data coming from the volatile profile characterization of biomass after fermentation. After the process, no change in the sample distribution can be observed. CHL1 forms a single cluster with CHL3, characterized by the presence of pyrazines and aromatic hydrocarbons. After fermentation, CHL2 biomass is now characterized by furans, esters, terpenes, acids, and sulfur-containing compounds, while fermented CHL4 is characterized by ketones hydrocarbons, aldehydes, and alcohols.

A total of 25 different volatile compounds were identified in CHL1 before fermentation. Its composition was characterized by two acids, five alcohols, four aldehydes, one ester, one furan, one aromatic hydrocarbon, two ketones, three pyrazines, and six terpenes (Appendix A). The dominant class of VOCs in this biomass was that of pyrazines. The fermentation processes with the different LAB have caused a reduction of VOCs in this biomass.

As can be seen from Figure 5a, in fermented CHL1, a significant decrease in aldehydes after fermentation with all the LAB strains was observed (*p* < 0.0001). Very interestingly, a complete reduction of hexanal has been noticed in all the fermented biomasses (Appendix A). Also, a significant reduction of acids has been seen after fermentation with *Lcb. rhamnosus* UPCCO 1473 (*p* < 0.0096), *L. bulgaricus* UPCCO 1932 (*p* < 0.0001), and *Lcb. casei* UPCCO 2240 (*p* = 0062). Fermentation has also caused a significant increase in alcohols (*p*< 0.0001). An augmented concentration of esters was observed when *Lcb. casei* UPCCO 2240 (*p*< 0.0001) and *Lcb. paracasei* UPCCO 2333 (*p* = 0.0021) were used for the fermentation, while furan content was not influenced by this process. A significant increase in aromatic hydrocarbons was obtained after the fermentation with *Lcb. rhamnosus* UPCCO 1473 (*p* < 0.0001), *L. bulgaricus* UPCCO 1932 (*p* < 0.0001), *Lcb. casei* UPCCO 2240 (*p* < 0.0001), *Lcb. paracasei* UPCCO 2333 (*p*< 0.0209), and *Leuc. citreum* UPCCO 4516 (*p* < 0.0001). A significant reduction in the presence of ketones was seen after the fermentation with *Lcb.casei* UPCCO 2240 (*p* < 0.0003), *Lcb. paracasei* UPCCO 2333 (*p* < 0.005), *Leuc. citreum* UPCCO 4516 (*p* < 0.0001), and *Lpb. plantarum* UPCCO 4932 (*p* < 0.0003). Furthermore, fermentation with all the LAB strains has significantly reduced the presence of pyrazines (*p* < 0.0001). A significant reduction of terpenes has been seen after fermentation with *L. bulgaricus* UPCCO 1932 (*p* < 0.0077).

CHL2 has proved to be the richest in volatile compounds. A total of 31 different VOCs were detected in the unfermented biomass, characterized by four acids, six alcohols, three aldehydes, one ester, two furans, one aromatic hydrocarbon, one hydrocarbon, seven ketones, and six terpenes. Figure 5b represents the impact of fermentation on microalgal biomass. Also, in this case, some strains reduced the amount of volatile compounds (*Lcb. casei* UPCCO 2240, *Lcb. rhamnosus* UPCCO 1473, and *Lcb. paracasei* UPCCO 2333). The fermentation process with all the selected LABs has significantly increased the presence of acids (*p* < 0.0001). After fermentation with *Leuc. citreum* UPCCO 4516 and *Lpb. plantarum* UPCCO 4932, a significantly higher presence of alcohols was detected (*p* < 0.0001), while UPCCO 2333 reduced their amount (*p* = 0.0133). A high quantity of aldehydes was detected in CHL2. All LAB have been able to significantly decrease these compounds (UPCCO 1473, UPCCO 2240, UPCCO 2333, and UPCCO 4932; *p* < 0.0001), and a complete reduction of hexanal was noticed in most of the fermented samples (exception for UPCCO 1932). Very interestingly the fermentations with *Lcb. paracasei* UPCCO 1473 (*p* < 0.0001), *Lcb. paracasei* UPCCO 2333 (*p* = 0.0001), *Leu. citreum* UPCCO 4516 (*p* < 0.0001), and *Lpb. plantarum* UPCCO 4932 (*p* < 0.0001) showed a significant increase in esters, which are responsible for fruity notes. Furans have been significantly reduced after fermentation with all lactic acid bacteria. A significantly reduced content of ketones was observed after fermentation with UPCCO 1473 (*p* < 0.0001), UPCCO 2240 (*p* = 0.043), UPCCO 2333 (*p* < 0.0001), UPCCO 4516 (*p* < 0.0001), and UPCCO 4932 (*p* < 0.0001). On the other hand, four out of six fermented samples (UPCCO 1473, UPCCO 2333, UPCCO 4516, and UPCCO 4932) showed the production of sulfur-containing compounds. A significant reduction of terpenes was revealed after fermentation with UPCCO 1473 (*p* < 0.0001), UPCCO 2333 (*p* < 0.0004), UPCCO 4516 (*p* < 0.001), and UPCCO 4932 (*p* < 0.0236).

A total of 24 VOCs were detected in CHL3, which was the *Chlorella* grown in complete heterotrophy and, for that reason, characterized by a negligible presence of chlorophylls. This biomass was characterized by two acids, four alcohols, eight different aldehydes, one ester, two aromatic hydrocarbons, four ketones, two pyrazines, and one terpene. The main class of volatiles observed in this biomass was that of aldehydes (Figure 5c). These components underwent a significant reduction (*p* < 0.0001) after the fermentation with the tested LAB. LAB strains have proved different behaviors regarding the production of acids. A significant increase in acids was observed after fermentation with UPCCO 1473, UPCCO 2240, and UPCCO 4932 (*p* < 0.0001), while UPCCO 2333 and UPCCO 4516 have reduced the presence of these compounds (*p* < 0.0001). The presence of esters was observed in the sample inoculated and fermented by *Leu. citreum* UPCCO 4516 (*p* = 0.0162). In the biomass fermented with UPCCO 4932, the aromatic hydrocarbons amount increased significantly (*p* < 0.0012). A significant reduction of ketones (*p* < 0.0001) and pyrazines (*p* < 0.0001) was revealed after fermentation with all the tested strains.

As mentioned before, CHL4 is the biomass with the highest presence of some classes of VOCs (aldehydes, alcohols, and ketones) (Figure 5d). However, this biomass is composed of a lower variability of compounds compared to the other *C. vulgaris* biomasses. In particular, 25 different VOCs were identified. CHL4 biomass was characterized by two acids, three alcohols, seven aldehydes, three ester, one hydrocarbon, and nine ketones. The statistical analysis has underlined how the fermentation process performed by UPCCO 1473 (*p* < 0.0077), UPCCO 2333 (*p* < 0.0001), and UPCCO 4516 (*p* < 0.0001) significantly reduced the quantity of aldehydes. A significant reduction of ketones was also revealed after fermentation with UPCCO 1473 (*p* < 0.0001), UPCCO 4516 (*p* < 0.0001), and UPCCO 4932 (*p* < 0.001).

## 4. Discussion

A growing tendency has been seen in the fermentation of microalgal biomass to obtain bioactive compounds or enhanced food products. However, the use of lactic acid fermentation as a tool to improve the volatile compounds of microalgae is still underexplored, and given its great potential, it is worth deepening the subject. In this work, four differently cultivated *C. vulgaris* biomasses were fermented with six different LAB species. Solid-state fermentation of *C. vulgaris* has been already applied to enhance the bioactivity of this Chlorophyta with very promising results [16]. In that study, four different species of LAB (*Lcb. rhamnosus*, *Lcb casei*, *Lb. delbrueckii bulgaricus*, and *Lcb. paracasei*) were used to ferment one strain of *C*. *vulgaris* (Nutrisslim, China) showing results similar to those obtained in the present work. The fermentability of this species of Chlorophyta by *Lcb. paracasei* UPCCO 2240, *Lcb. casei* UPCCO 1473, and *L. delbrueckii* UPCCO 1932, despite the different composition of the biomass, has been confirmed. However, the variability in fermentation capability exhibited by the same LAB when fermenting algae with different compositions must be underlined. This confirms the importance of selecting the appropriate starter culture for the production of novel fermented foods, which are of growing interest for their innovative characteristics and the great beneficial potential on consumers’ health [19,21]. Moreover, the results obtained in this study underline the validity of solid-state fermentation as a tool to ferment microalgae. Despite being currently underexplored, this approach appears promising. The solid-state fermentation has been previously applied to other species of algae, like *A*. *platensis* and *Chlorococcum*, showing a good ability of LAB to grow on these substrates. Furthermore, *Chlorella vulgaris* biomass has been studied as a supplement to increase the growth of LAB [22,23]. In those studies, supplementing media or milk with 0.1%, 1%, or 1.5% of this microalga has been shown to enhance the survivability and growth of LAB strains. Similar results were obtained with other algae, suggesting the positive impact of the high protein and carbohydrate content of microalgae on LAB growth [15]. In this study, all *C. vulgaris* biomasses served as excellent substrates for lactic acid fermentation, though differences were observed in the growth capabilities of various tested LAB strains across different biomasses. These differences could be attributed to the distinct compositions of the microalgae in protein, carbohydrates, fibers, and lipids. Among them, CHL2, the biomass with the highest protein content, was the most suitable for fermentation. Similarly, *A. platensis*, another protein-rich biomass, has demonstrated excellent fermentability [17,24].

Previous studies have been conducted on the characterization of the aromatic fraction of microalgae, such as *C. vulgaris* [10,25,26,27]. However, as far as the authors know, few works are present in the literature analyzing the volatile compounds of heterotrophically cultivated *C. vulgaris* [28]. Our results strengthen the knowledge on this topic, revealing different VOC compositions depending on the biomass and the growth conditions. Building on this, the VOC composition of the algal biomasses was analyzed following LAB solid-state fermentation, as a key objective of this study was to assess the impact of LAB fermentation on the VOC profile of *C. vulgaris* biomasses. These biomasses are known for their unpleasant odor, which often limits their use as a food ingredient. Previous studies have explored solid-state fermentation as a potential solution to this issue [17]. Fermentation with *Lcb*. *casei* UPCCO 2240 was previously shown to have a good impact on the volatile compound composition of *A*. *platensis* biomass. A reduction of aldehydes has been observed after fermentation with this strain, probably due to the contemporary formation of alcohols via reduction mechanisms during fermentation [29]. Similarly to what was observed in this study, lactic acid fermentation was capable of reducing hexanal with a consequent decrease in the green and leafy aroma, seldom undesired in food matrices. The reduction of off-flavors of microalgae is generally achieved by applying an extraction procedure with a solvent like ethanol, acetone, or hexane to dissolve the aromatic compounds [30]. However, this method has some limitations from both the economic and ecological point of view [31]. Conversely, the potential to utilize an eco-friendly, sustainable, and straightforward process like fermentation is highly appealing and promising for both producers and consumers. The fermentation of microalgae with LAB has previously been shown to effectively eliminate volatile compounds in algal biomass while generating new components, such as acetoin, which contribute to fermented and creamy aromatic notes [17,32]. In the present study, different LAB strains, belonging to different species, were used to tackle off-flavors of different *C. vulgaris* via fermentation, and changes in the volatile fraction were determined.

In general, the dominant classes of VOCs in the analyzed biomasses, both fermented and non-fermented, were acids, alcohols, aldehydes, and ketones. Different organic acids were detected in the starting materials, and the fermentation process led to the production of and/or increase in organic acids, both in terms of diversity and quantity. These components presented higher concentrations in CHL2 samples, especially after the fermentation step, and they contributed to differentiating these samples from the others.

Alcohols and aldehydes may be present because they are generated from the oxidative degradation of unsaturated fatty acids [33]. Linear aldehydes, both saturated and unsaturated, containing from 6 to 9 carbon atoms as hexanal, heptanal, octanal, nonanal, 2-octenal, etc., have been associated with green aromatic notes, so they can contribute to the typical “seaweed” flavor [34,35]. The fermentation step may serve to decrease the concentration of these components, as observed in a present study, contributing to diminishing compounds that can give unpleasant notes and improving the aromatic profile of *C. vulgaris*. Among alcohols, particularly interesting is 1-octen-3-ol for its mushroom-like odor. This compound has been already detected in *Chlorella* and other microalgae species [26,35]. When it comes to aldehydes, again, fermentation may contribute to reducing the amount of 1-octen-3-ol, enhancing the product’s sensory profile and flavor. In CHL4, ketones were also detected in higher quantities compared to other fermented biomasses. Ketones were found to be the most representative volatile compounds in different microalgae species, such as *A. platensis*, *Chlorella pyrenoidosa*, *Chlamydomonas reinhardtii*, and *Haematococcus pluvialis* [35].

Other volatile compounds revealed in fermented CHL2 and CHL4 were esters. The presence of esters is seldom connected to sweet and fruity aroma in foods [36], thus suggesting the role of fermentation in improving the aroma of *Chlorella* biomasses. On the other hand, these compounds were also found in a low amount in non-fermented *Chlorella* (CHL2, CHL3, and CHL4), in agreement with Zhao et al. 2024 [35].

Pyrazines were detected in CHL1 and CHL3 before and after the fermentation step. This class of compounds is responsible for clustering and differentiation of these samples from the others. Pyrazines are associated with nutty and roasted aromatic notes present in heat-treated matrices [37]. The amount of pyrazine was influenced by the fermentation process, which led to a decrease in their quantity both in CHL1 and CHL3. This behavior was observed also by Zhang and collaborators after the fermentation of *Chlorella pyrenoidosa* by *Bacillus velezensis* [38].

Among furans, 2-pentyl furan was detected in almost all the analyzed samples. This compound is associated with fruit, earthy, green, and vegetable-like notes, and it can be considered a key odorant in algae aroma [39]. Overall, the concentration of furans was decreased by the fermentation step.

The increase in sulfur-containing compounds can be linked to the presence of sulfuric amino acids in the biomass. These compounds were detected in CHL2, especially after the fermentation step, and in particular, dimethyl disulfide and dimethyl trisulfide were identified. This compound was recently identified by Zhao et al., who analyzed the composition of various microalgae species, highlighting it as a characteristic volatile compound associated with different types of microalgal products [35]. Finally, some compounds pertaining to the categories of hydrocarbons and terpenes were identified in almost all the analyzed samples. Though in a lower amount compared to other chemical classes, these components contributed to differentiating the biomasses, both before and after fermentation. However, regardless of different hydrocarbons and terpenes found in the analyzed samples, their contribution to the final aroma of microalgae may be neglected; in particular, the odor threshold of hydrocarbons is indeed very high, so at the level detected in this study, they would probably not be perceived by a potential consumer [35].

## 5. Conclusions

A significant barrier to the broader acceptance of microalgal food products is the presence of distinct off-flavors. Our findings provide compelling evidence that solid-state lactic acid fermentation offers a promising solution to this challenge.

All the tested *C. vulgaris* biomasses, despite a varying composition, demonstrated their suitability as matrices for the growth of LAB. Notably, fermentation enhanced the VOC profiles of the fermented biomasses, effectively reducing undesirable flavors such as hexanal and other VOCs related to their green and fishy aroma. Among the evaluated LAB strains, several emerged as strong candidates serving as starter cultures to deodorize algal biomasses. Particularly, *Lcb. paracasei* UPCCO 2333 was demonstrated to have the most favorable impact on the volatile compound profile of *Chlorella vulgaris* biomasses. It consistently reduced undesirable compounds, such as aldehydes, ketones, pyrazines, and terpenes, across all biomass types, while significantly enhancing the presence of esters, key contributors to fruity and pleasant aromas. Unlike some other strains, UPCCO 2333 did not produce sulfur-containing compounds, which are often associated with off-flavors. These results suggest that *Lcb. paracasei* UPCCO 2333 is the most effective strain for improving the sensory quality of microalgal biomass through fermentation. These results pave the way for innovative applications of microalgae as versatile and appealing food ingredients.

## Figures and Tables

**Figure 1 foods-14-01511-f001:**
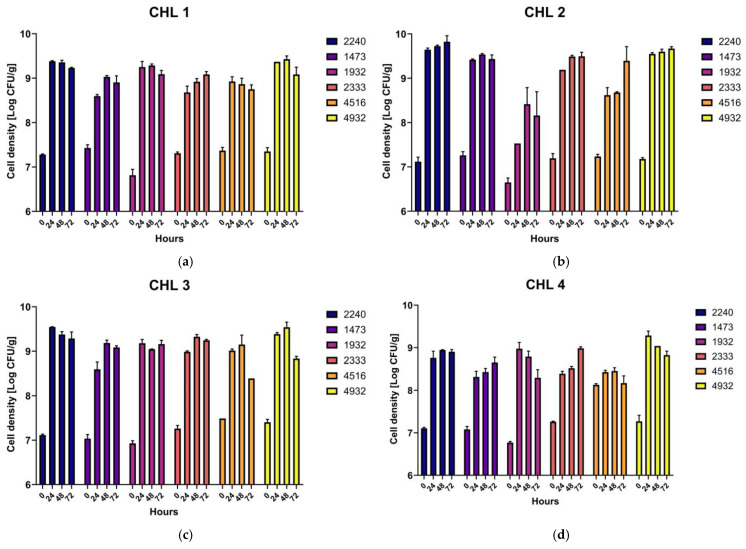
Growth curves of LAB during fermentations of *Chlorella vulgaris* biomasses. The microbial concentration was evaluated just after the inoculum (0), and after 24 h (T1), 48 h (T2), and 72 h (T3) of fermentation. In blue, *Lacticaseibacillus casei* UPCCO 2240; in purple, *Lacticaseibacillus rhamnosus* UPCCO 1473; in pink, *Lactobacillus delbrueckii bulgaricus* UPCCO 1932; in red, *Lacticaseibacillus paracasei* UPCCO 2333; in orange, *Leuconostoc citreum* UPCCO 4516; in yellow, *Lactiplantibacillus plantarum* UPCCO 4932. (**a**) CHL1, (**b**) CHL2, (**c**) CHL3, and (**d**) CHL4.

**Figure 2 foods-14-01511-f002:**
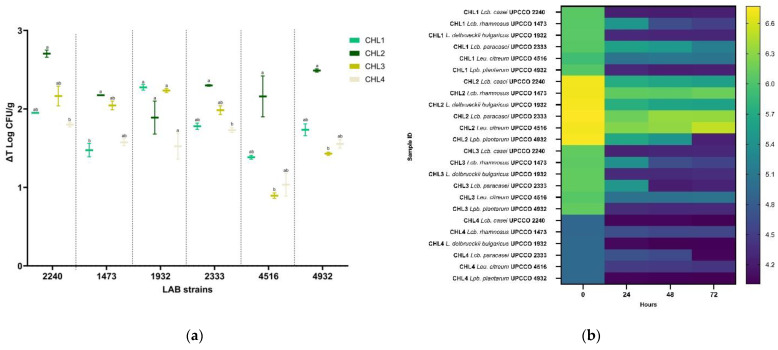
(**a**) Boxplot picturing the ΔT = T3-T0 of LAB strains growth in *C. vulgaris* samples. Comparisons between the strain’s ability to ferment different matrices are expressed with letters. (**b**) Heatmap representing pH kinetics during fermentations of *C. vulgaris* biomasses. The pH value was measured just after the inoculum (0), and after 24 h (T1), 48 h (T2), and 72 h (T3) of fermentation. LAB strains: *Lacticaseibacillus casei* UPCCO 2240, *Lacticaseibacillus rhamnosus* UPCCO 1473, *Lactobacillus delbrueckii bulgaricus* UPCCO 1932, *Lacticaseibacillus paracasei* UPCCO 2333, *Leuconostoc citreum* UPCCO 4516, and *Lactiplantibacillus plantarum* UPCCO 4932.

**Figure 3 foods-14-01511-f003:**
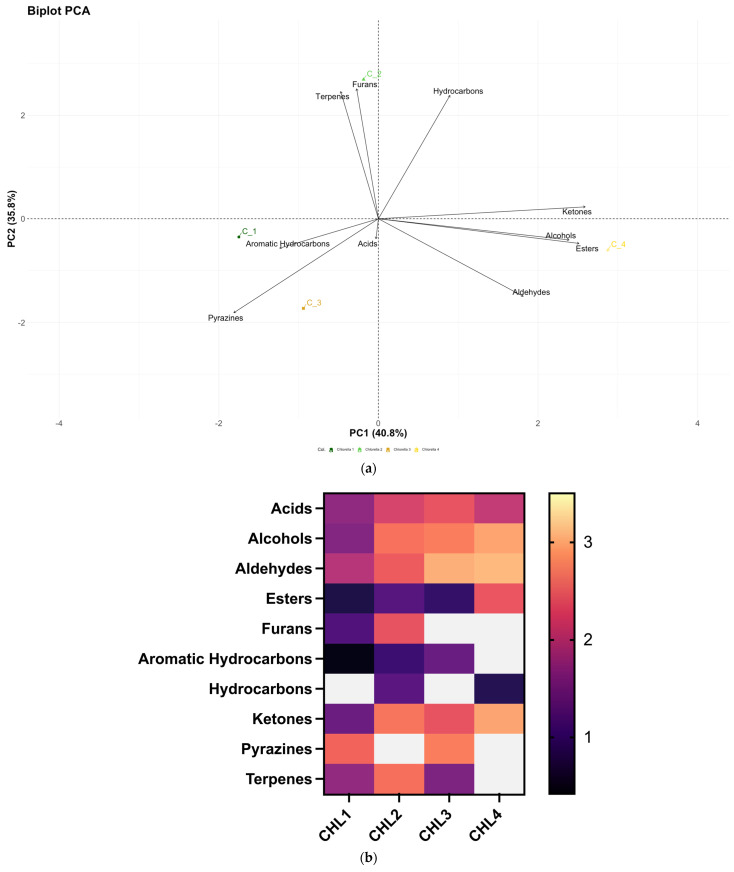
(**a**) PCA biplot built considering as variables the concentrations of the different VOCs classes identified in the non-fermented *C. vulgaris* samples. In dark green CHL1(C1), light green CHL2 (C2), orange CHL3 (C3), and in yellow CHL4 (C4). (**b**) Heatmap representing the classes of the 4 different not fermented *C. vulgaris* biomasses (Log ng/g).

**Figure 4 foods-14-01511-f004:**
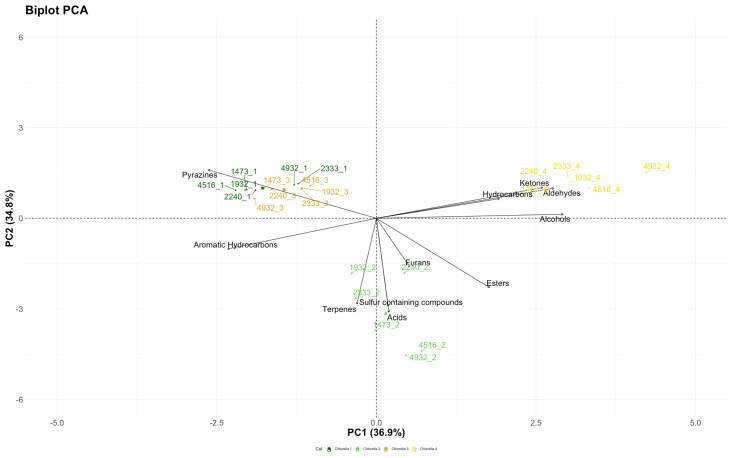
PCA biplot built considering as variables the concentrations of the different VOCs classes identified in fermented *C. vulgaris* samples. In dark green, CHL1; light green, CHL2; orange, CHL3; and yellow, CHL4.

**Figure 5 foods-14-01511-f005:**
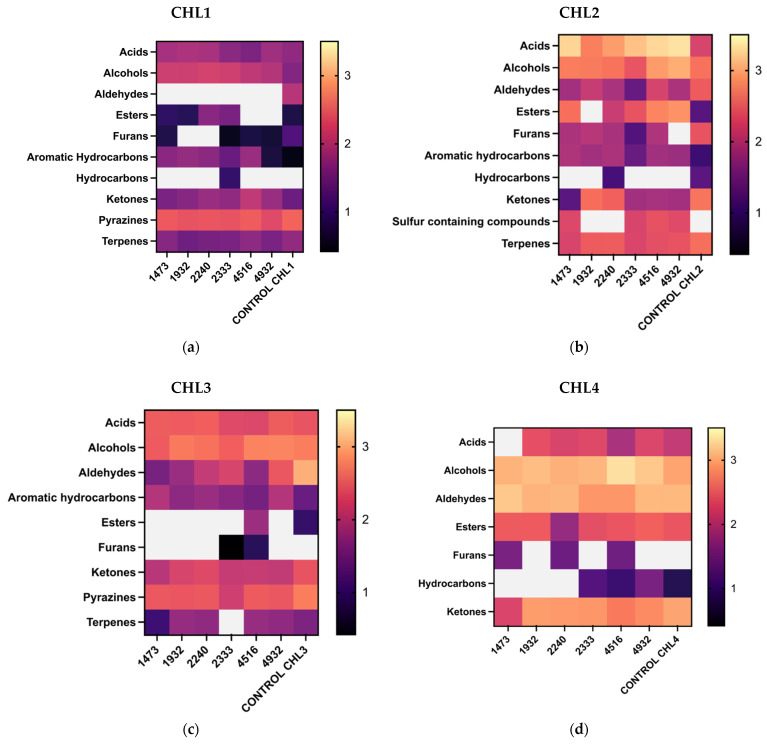
Heatmaps representing the classes of the 4 different fermented *Chlorella vulgaris* biomasses (Log10 ng/g): (**a**) CHL1, (**b**) CHL2, (**c**) CHL3, and (**d**) CHL4.

**Table 1 foods-14-01511-t001:** Nutritional composition of commercial *Chlorella vulgaris* biomasses (%) provided by the Allmicroalgae (Leira, Portugal).

Name	Carbohydrates	Proteins	Fat	Fibre
*C. vulgaris* “Smooth” CHL1	34	30	7	20
*C. vulgaris* “Premium” CHL2	8	55	10	15
*C. vulgaris* “Honey” CHL3	24	30	8	23
*C. vulgaris* “White” CHL4	55	32.5	9.5	9

**Table 2 foods-14-01511-t002:** LAB used for the fermentation of microalgal biomasses.

ID	Species	Growth Temperature	Isolation Matrix
UPCCO 1473	*Lacticaseibacillus rhamnosus*	37 °C	Parmigiano Reggiano
UPCCO 1932	*Lactobacillus delbrueckii bulgaricus*	37 °C	Curd
UPCCO 2240	*Lacticaseibacillus casei*	37 °C	Parmigiano Reggiano
UPCCO 2333	*Lacticaseibacillus paracasei*	30 °C	Parmigiano Reggiano
UPCCO 4516	*Leuconostoc citreum*	30 °C	Sourdough
UPCCO 4932	*Lactiplantibacillus plantarum*	30 °C	Minas cheese

**Table 3 foods-14-01511-t003:** Growth ability of different LAB species/strains on four different *C. vulgaris* biomasses after 72 h of fermentation at the optimal growth temperature (37 and 30 °C). Values are reported as Log CFU/g ± standard deviation. ΔT = T3-T0 represent microbial increase after 72 h.

	*CHL 1*			*CHL 2*			*CHL 3*			*CHL 4*		
ID	T0	T3	ΔT = T3-T0	T0	T3	ΔT = T3-T0	T0	T3	ΔT = T3-T0	T0	T3	ΔT = T3-T0
UPCCO 1473	7.43 ± 0.07	8.90 ± 0.15	1.48 ± 0.12	7.26 ± 0.08	9.44 ± 0.09	2.17 ± 0.01	7.04 ± 0.09	9.09 ± 0.04	2.05 ± 0.08	7.08 ± 0.08	8.65 ± 0.12	1.57 ± 0.07
UPCCO 1932	6.82 ± 0.13	9.09 ± 0.09	2.27 ± 0.05	6.65 ± 0.10	8.54 ± 0.31	1.89 ± 0.30	6.93 ± 0.06	9.16 ± 0.08	2.23 ± 0.03	6.77 ± 0.03	8.30 ± 0.19	1.53 ± 0.23
UPCCO 2240	7.28 ± 0.01	9.23 ± 0.02	1.95 ± 0.01	7.12 ± 0.09	9.82 ± 0.14	2.71 ± 0.06	7.12 ± 0.02	9.29 ± 0.15	2.17 ± 0.18	7.11 ± 0.03	8.91 ± 0.05	1.80 ± 0.04
UPCCO 2333	7.31 ± 0.03	9.09 ± 0.07	1.78 ± 0.06	7.20 ± 0.10	9.50 ± 0.09	2.30 ± 0.01	7.26 ± 0.08	9.24 ± 0.02	1.98 ± 0.08	7.26 ± 0.01	8.99 ± 0.04	1.73 ± 0.04
UPCCO 4516	7.35 ± 0.08	8.75 ± 0.10	1.38 ± 0.04	7.24 ± 0.05	9.40 ± 0.32	2.16 ± 0.37	7.49 ± 0.06	8.39 ± 0.10	0.90 ± 0.05	8.13 ± 0.02	8.17 ± 0.17	1.04 ± 0.21
UPCCO 4932	7.37 ± 0.08	9.15 ± 0.16	1.73 ± 0.11	7.18 ± 0.02	9.67 ± 0.04	2.49 ± 0.03	7.41 ± 0.07	8.84 ± 0.05	1.43 ± 0.05	7.27 ± 0.14	8.83 ± 0.09	1.55 ± 0.08

## Data Availability

The original contributions presented in the study are included in the article/Appendix A, further inquiries can be directed to the corresponding author.

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
