# Peer review of "Lactic Acid Fermentation of Chlorella vulgaris to Improve the Aroma of New Microalgae-Based Foods: Impact of Composition and Bacterial Growth on the Volatile Fraction"

_foods, 2025, doi:10.3390/foods14091511_

Round 1

Reviewer 1 Report

Comments and Suggestions for Authors

I read the manuscript 'Lactic acid fermentation of Chlorella vulgaris to improve the aroma of new food products: impact of composition and bacterial growth on the volatile fraction', evaluating the impact of lactic acid fermentation on the aromatic profiles of four distinct Chlorella vulgaris biomasses. The introduction is not too long, but sufficient. The research is well planned. The statistical analyses performed are at the good level. Quite detailed discussion of results. 

Table 1 - are these results obtained by you, or given by the biomass distributor, or maybe from some publications - citation needed.

Unfortunately, I do not have access to the supplementary materials, which is the basis for all statistical analyses, so it is difficult for me to comment on the correctness of the conclusions obtained.

Is it possible to draw results corresponding to the title of the work, i.e. "improve the aroma", based on the obtained results? I have concerns about this. The optimal solution here would be to create the sensory panel, and the minimum version would be to show changes taking into account the aroma of key compounds, their thresholds of detection and the strength of the impact on the overall aroma.

Good selection of literature. I checked the titles I had access to and didn't notice anything incorrect.

Tables and figures very well developed. I really appreciate such visually good statistical studies.

Please attach the supplement so I can comment on the analysis results.

Author Response

I read the manuscript 'Lactic acid fermentation of Chlorella vulgaris to improve the aroma of new food products: impact of composition and bacterial growth on the volatile fraction', evaluating the impact of lactic acid fermentation on the aromatic profiles of four distinct Chlorella vulgaris biomasses. The introduction is not too long, but sufficient. The research is well planned. The statistical analyses performed are at the good level. Quite detailed discussion of results.

Table 1 - are these results obtained by you, or given by the biomass distributor, or maybe from some publications - citation needed.

Thank you very much for the observation, the compositional data are provided by the producer. We have added it in the caption of the table.

Unfortunately, I do not have access to the supplementary materials, which is the basis for all statistical analyses, so it is difficult for me to comment on the correctness of the conclusions obtained.

Is it possible to draw results corresponding to the title of the work, i.e. "improve the aroma", based on the obtained results? I have concerns about this. The optimal solution here would be to create the sensory panel, and the minimum version would be to show changes taking into account the aroma of key compounds, their thresholds of detection and the strength of the impact on the overall aroma.

We agree that conducting a panel test would be a valuable approach to more effectively assess human perception of the aroma improvement. Aroma formation depends on a complex mixture of various chemical groups, making the analysis of these compounds particularly challenging. Typically, four main steps are involved in the analysis of aroma compounds: isolation, separation, identification, and sensory evaluation (Al-Khalili, P. et al., Aroma compounds in food: Analysis, characterization and flavor perception, Measurement: Food, Volume 18, 2025, https://doi.org/10.1016/j.meafoo.2025.100220). Using gas chromatography combined with mass spectrometry, which is the most widely used method for separating and identifying volatile compounds, we described the impact of LAB fermentation on the compounds responsible for aroma formation. Since the goal of our study was to evaluate the potential of fermenting microalgal biomass for the development of new food products, sensory evaluation of the final products will certainly be included in future research.

Good selection of literature. I checked the titles I had access to and didn't notice anything incorrect.

Tables and figures very well developed. I really appreciate such visually good statistical studies.

Please attach the supplement so I can comment on the analysis results.

We apologize for the inconvenience. We checked the link to Zenodo where the material is deposited, and it is actually working. So please try again and find the Supplementary File available at the following link:

https://zenodo.org/records/15234687?preview=1&token=eyJhbGciOiJIUzUxMiJ9.eyJpZCI6ImQ5YTMxMThhLTgxZDQtNDkzNy04MzZjLTE4ZmI2MTdlNGYxZCIsImRhdGEiOnt9LCJyYW5kb20iOiIxMTRiN2E1Y2M0MDM4NzJkY2FlMGI0YzgxYmMxOTRjZCJ9.j01Y4z19yB-qHn3vnXE2bPrwWeTWClGufogbd6fayy31o-Xwi9n-LGZEoZg7nB4Ejcfrx2LyF3HjTgurLx8kSg

Reviewer 2 Report

Comments and Suggestions for Authors

Dear Author,

Your manuscript makes a valuable contribution to the improvement of aroma in novel food products. The research is of interest and suitable for publication in Foods; however, there are a few concerns within the text that should be addressed prior to publication. My specific comments are outlined below. In my opinion, the article needs minor revisions.

  • The term new food products in the title gives the impression that many new food products are studied in the article. However, the study is about microalgae-based foods. Instead of new food products in title please use microalgae-based foods.
  • The abstract should include specific data on the aroma compounds identified in this study to provide a clearer summary of the key findings.
  • line 48-50: please give additional information about production of volatile organic compounds (VOCs) by microalgae under common conditions.
  • line 113 and line 121: log cfu/g or mL ! Which one is true for biomass? please use the same style.
  • line 154: be careful about the size of table 3
  • line 287: figure 5 needs figure legend or caption
  • In the discussion section, please include data from previous studies to enable a comparison with the results of the present study. Specifically, data on aroma compounds under different fermentation conditions would allow for a more meaningful comparison and interpretation of the current findings.
  • The references throughout this manuscript are relevant; however, some formatting corrections are needed. Please italicize the names of microorganisms and ensure consistent formatting of journal titles—either use the full name or the standard abbreviation, but apply it uniformly.

Author Response

Dear Reviewer, 

Please find our responses to your observations in the attached Word file.

Best regards,

Reviewer 3 Report

Comments and Suggestions for Authors

Dear Authors,

In general, the manuscript is adequate to results presented in this paper the structure of the work is clear and complete. After reading the paper, it is clear that the authors have experience in fermentation of strains of Chlorella vulgaris and six lacidbaccilius, to have biomass with less volatile compounds of unpleasant odor. Nomenclature concerning on these processes is proper. It easy to found, that  one of authors is experience in this research area guarantee the quality of presented paper and the work is clear and looks complete. Chlorella vulgaris is a promising source of energy production, being a good alternative to biofuel crops, like soybean, corn or rapeseed, as it is more productive and does not compete with food production. It can produce large amount of lipids, up to 20 times more than crops. Microalgal biofuels are not competitive with fossil fuels, however, it is place to inform of other uses.  

The Authors performed a lot of experiments and analyzes in order to obtain detailed research results on fermentation of microalgae biomasses and the paper looks complete. However, see my comments below, which most of them should be helpful in a clarification to readers:

  1. Lines 102-103 – Table 2. - Why were used different temperature for LAB strains (UPCCO 1473, UPCCO 1932, and UPCCO 2240) that were incubated for 16 hours under aerobic conditions at 37°C and others strains at 30°C?
  2. Line 153 - Table 3 It should be ΔT=T3-T0 instead Δ (T3-T0). In the column it should be also ΔT instead Δ (T3-T0).
  3. Line 160 – Figure 1. The results should be presented as bars. You should not draw lines between studied points, because you have any information in the range between. In the same Figure 1. Microbial concentrations were assessed after inoculation time: (T0) after 24 hours (T1), after 48 hours (T2), and after 72 hours (T3) of fermentation. Why on Figure 1a time (T2) is after 36 hours, while in others (Fig. 1 b, c, d) is after 48 hours?
  4. Lines 466-477 – Conclusions. The conclusions are formulated to generally and tgere are any useful detail. It should be good if you indicate which of six lactic acid bacteria used in this study is most promised to reduce unpleasant volatile compounds? On the other hand, in abstract, you write that: “Fermentation notably reduced concentrations of compounds responsible for off-flavors, such as aldehydes. Specifically, hexanal, associated with a green and leafy aroma, was significantly decreased. These results highlight lactic acid fermentation as an effective method to improve the sensory characteristics of C. vulgaris biomasses, enabling their broader use in innovative, nutritionally rich food products”. It should be inserting in the conclusions, I guess.

I recommend it minor revision.

Author Response

(The authors gave the same response as above.)

Round 2

Reviewer 1 Report

Comments and Suggestions for Authors

Ethyl 3,3-dimethylbutyrate is an ester, not an acid.

On what basis were the exact enantiomers of only some compounds (cis-; (Z); (E); S-) determined? Especially in the case of 2,3-Butanediol, it seems strange to me that only for one sample was the exact enantiomer determined (2,3-Butanediol, [S-(R*,R*)]-). Please check the literature to see if the compounds you have determined match the literature data. Specific enantiomers are always very characteristic for biological samples.

Author Response

First of all, we thank the reviewer for this interesting discussion. A distinction must be made between positional isomers and enantiomers. In particular, the position of substituents in respect to a double bond (cis or trans; Z or E; etc.) can be deduced from the mass spectrum of the molecule, without the need for a reference standard or a particular stationary phase. With regard to isomers such as enantiomers, we agree with the reviewer that there is a need to compare with literature data, or with a standard, or to use a column with a chiral stationary phase. So in the case of molecules as 2,3-Butanediol, [S-(R*,R*)] actually we are not 100% sure of the identification. So we have substituted with "2,3-Butanediol". 

We are sorry for the mistake. Ethyl 3,3-dimethyl butyrate has been moved to the esters. All figures, tables and discussion in the manuscript have been updated considering the change.